# Fire-Resistant Sandwich-Structured Composite Material Based on Alternative Materials and Its Physical and Mechanical Properties

**DOI:** 10.3390/ma12091432

**Published:** 2019-05-02

**Authors:** Štěpán Hýsek, Miroslav Frydrych, Miroslav Herclík, Petr Louda, Ludmila Fridrichová, Su Le Van, Hiep Le Chi

**Affiliations:** 1Department of Material Science, Faculty of Mechanical Engineering, Technical University of Liberec, Studentska 2, 461 17 Liberec, Czech Republic; miroslav.frydrych@tul.cz (M.F.); miroslav.herclik@tul.cz (M.H.); petr.louda@tul.cz (P.L.); longsuvp90@gmail.com (S.L.V.); lechihieptul09@gmail.com (H.L.C.); 2Department of Textile Evaluation, Faculty of Textile Engineering, Technical University of Liberec, Studentska 2, 461 17 Liberec, Czech Republic; ludmila.fridrichova@tul.cz

**Keywords:** composite material, sandwich panel, rapeseed, geopolymer, reinforcing lattice

## Abstract

The development of composite materials from alternative raw materials, and the design of their properties for the intended purpose is an integral part of the rational management of raw materials and waste recycling. The submitted paper comprehensively assesses the physical and mechanical properties of sandwich composite material made from particles of winter rapeseed stalks, geopolymer and reinforcing basalt lattices. The developed composite panel is designed for use as a filler in constructions (building or building joinery). The observed properties were: bending characteristics, internal bonding, thermal conductivity coefficient and combustion characteristics. The results showed that the density of the particleboard has a significant effect on the resulting mechanical properties of the entire sandwich panel. On the contrary, the density of the second layer of the sandwich panel, geopolymer, did not have the same influence on its mechanical properties as the density of the particleboard. The basalt fibre reinforcement lattice positively affected the mechanical properties of sandwich composites only if it was sufficiently embedded in the structure of the particle board. All of the manufactured sandwich composites resisted flame for more than 13 min and the fire resistance was positively affected by the density of the geopolymer layer.

## 1. Introduction

Through their properties, basalt fibres are intended for use as a reinforcement in composite materials. These fibres have a higher tensile strength than E-glass fibres, larger strain to failure than carbon fibres and good resistance to alkaline exposure; they are also non-flammable, chemically stable, non-toxic and, overall, can be used in conditions from −200 °C to 600 °C [1,2,3]. Basalt fibres may be used separately as microfibers, successfully fulfilling the function of a reinforcing agent in composite materials based on inorganic matrices [4,5,6,7], or as a reinforcement element in the form of a surface fabric, the use of which appears to be very effective [1,8]. In addition, basalt fibres arranged in the form of a surface fabric have been successfully used in the past, for example in reinforcing concrete beams [9], historical pillars [10] or surface panels [1,11,12]. In all of the mentioned cases, basalt fibres, whether as single fibres, fibre bundles or fibre lattices, function as a reinforcement element in a matrix made of concrete, mortar or other material. In the presented research, the basalt fibre surface fabric was used to reinforce the particleboard from winter rapeseed stalks.

Winter rapeseed stalks are an abundant raw material in the European Union, with about 45 million tonnes produced annually [13], which would approximately two times suffice for the annual consumption of wood for the production of particleboard in the EU [14]. In addition to the production of bioethanol [15], it can be successfully used in the production of particleboard [16].

In terms of carbon dioxide binding, their use for further production of material is far more meaningful than their use for energy purposes. Boards made from these stems can also have very good properties [17]. However, the combination of these post-harvest residues with cheap urea-formaldehyde adhesive is very attractive, which offers a very cost-effective composite material with acceptable properties [18]. Geopolymer was used in order to improve the mechanical properties and fire resistance of this cheap composite reinforced with basalt surface fabric.

Geopolymers are amorphous three-dimensional alkali-activated aluminosilicates [19] and are considered an environmentally friendly building material [20]. These are materials that have great potential in various applications due to their specific properties (thermal insulation, fire resistance, strength, acoustics). Due to the possible foaming of the geopolymer, a lightweight material is produced while maintaining the fire and strength characteristics. Examples include a geopolymer-based composite and basalt fibre, which is suitable for high-temperature applications [4] or geopolymer foam concrete (GFC) panel with excellent sound absorption [21].

The sustainable development of production, processing and consumption cannot be achieved without the responsible waste management [22] and efficient use of materials [23,24]. This paper responds to the latest research trends and deals with the development of a new lightweight composite material from alternative raw materials for the production of highly functional composite materials with properties for the intended use. The aim of the work is to determine the influence of density of individual layers of sandwich composite material and reinforcing lattices on its mechanical and physical properties.

## 2. Materials and Methods

Particles from winter rape stalks were used to produce particleboards. The producer of particles from winter rape stalks was Mikó Stroh (Borota, Hungary). The dimensional characteristics of the used particles are given in Table 1. The dimensional characteristics of particles were determined using screen analysis (Imal, Modena, Italy).

### 2.1. Lattice

Lattices made from basalt fibre with loop dimensions of 25 mm were supplied by Alligard (Libavské Údolí, Czech Republic). According to the producer’s declaration from the technical data sheet, each bundle of fibres contained 6500 microfibers with a diameter of 11–18 µm. The linear mass density of the bundle was 2400 tex and the density of microfibers was 2700 kg/m^3^. The tensile strength of the individual bundles was 600 MPa. According to the variants, two, one or no lattices were pressed into the particleboard. In the composite material with the two-lattice variant, the lattices were placed in the surface layer, and in the variant with one lattice it was pressed in the middle of the board. The lattices were inserted into the particles during the layering of the particle carpet, according to the variant in the middle of the carpet, or on its surface from both sides (Figure 1).

### 2.2. Adhesive Mixture Application

A urea-formaldehyde adhesive (UF) was used to manufacture particleboards, which was applied to particles using a laboratory adhesive applicator. The used hardener was (NH_4_)(NO_3_), and the ratio between the solids hardener and dry adhesive was 10%. The hydrophobizing agent used was paraffin emulsion. The solid content of whole adhesive mixture was 50%. A resin dosage of 10% solids on particle dry mass was used. The detailed composition of the adhesive is given in [25].

### 2.3. Pre-Pressing and Hot-Pressing

Particleboards were pressed from particles dried to 8% moisture content. The first step was cold pre-pressing with conditions: pressure 4 bars, time 1 min. The pre-pressed board was pressed using a heated HLP350 hydraulic press (Höfer Presstechnik GmbH, Taiskirchen, Austria) at 165 °C. Pressing was carried out according to the press cycle (Table 2). The nominal board thickness was 12 mm. Plates were pressed in two variants according to density and in three variants according to the number of grids. An illustration of the various composite variants is given in Table 3. After pressing, the boards were conditioned at 20 °C and at a relative humidity (RH) of 65% until moisture stabilization.

### 2.4. Geopolymer Application

In order to increase the fire resistance of the manufactured materials, geopolymer was applied to the manufactured board from one side with a nominal height of the layer of 1 cm and a nominal density of 885 and 915 kg/m^3^. The variants of the manufactured sandwich-structured panel are listed in Table 3. In order to compare the developed material with commercially-sold products, a commercially-sold oriented strand board (OSB) (class 3, density 620 kg/m^3^) was selected. This board was used for reference samples instead of particleboard from rapeseed stalks and reinforced lattices. Both geopolymer variants were also applied to the OSB.

The mixture for the geopolymer production consists of the following five components.
(1)inorganic, two-component, aluminosilicate binder based on metakaolin Cement Baucis Lk (České Lupkové Závody, a.s., Nové Strašecí, Czech Republic),(2)alkaline activator in liquid form Activator Baucis Lk (České Lupkové Závody, a.s., Nové Strašecí, Czech Republic),(3)anticorrosive powder additive for concrete and malt based on amorphous SiO_2_ Kema Mikrosilika (Kema Mikrosilika-sanační centrum s.r.o., Sviadnov Czech Republic),(4)basalt fibres Mineral wool Isover Uni—basalt fibres (Saint-Gobain Construction Product CZ a.s., Praha, Czech Republic),(5)aluminium powder with a purity of at least 99% and an average particle size of 65 µm Aluminium powder-Alpra—very fine, (PK Chemie, Třebíč Czech Republic). The geopolymer was manufactured according the methodology previously published in [8]. Two manufactured geopolymer density variants were selected; the percentage of all components in each variant is shown in Table 4.

### 2.5. Composite Material Properties Estimation

Before all of the tests, the panels we air-conditioned at 20 °C and a relative humidity (RH) of 65% for three weeks. After this period, the equilibrium moisture was achieved. The density, the tensile strength of the composite material perpendicular to the board’s plane and the three-point bending characteristics were measured according international standards. The density of boards was measured according to EN 323 [26], the internal bonding was measured according to EN 319 [27] and the measurements of three-point bending characteristics were carried out according to EN 798 [28]. The bending test bodies were placed on the universal testing machine so that the geopolymer layer was directed upward in order to be subjected to compressive stress during bending, and the layer with reinforcing lattice to tensile stress. The measurement accuracy of the universal tensile machine was 0.25% of reading. For a more thorough characterization of the bending properties of sandwich composites, the bending coefficient was calculated according to the following Formula [29]:
(1)$$K_{bendC}=\frac{h}{R_{minC}}=\frac{h}{\frac{l_{0}^{2}}{12\cdot y_{max}}}$$
where:
$R_{minC}$—The minimum curve radius based on the basic bending equations.$K_{bendC}$—The coefficient of bendability based on the basic bending equations.$y_{max}$—The maximum deflection.$l_{0}$—The distance between supports.*h*—The thickness of the sample.

The tensile and bending tests were performed using universal testing machine TIRA test 2850. The thermal conductivity of boards was estimated using Isomet 2104 (Applied Precision, Ltd., Bratislava, Slovakia) according the method described previously in [30], the thermal conductivity was measured using a probe with a range of 0.015–2 W/(m·K). According to the technical data specification of the probe, the measurement accuracy is 5% of reading +0.001 W/(m·K). The thermal conductivity coefficient of the entire sandwich panel was subsequently calculated. The recalculation was performed according to the thermal resistances of the individual layers according to Formula (2). The total coefficient of thermal conductivity of the developed sandwich panels is therefore the theoretical value based on the values of the thermal resistances of the individual layers and does not include thermal resistance during heat transfer.
(2)$$\lambda_{tot}=\frac{d_{tot}}{\sum R_{i}}=\frac{d_{tot}}{\sum \frac{d_{i}}{\lambda_{i}}}$$
where:
$\lambda_{tot}$—The total thermal conductivity of the sandwich panel.$d_{tot}$—The total thickness of the sandwich panel.$d_{i}$—The thickness of one layer in the sandwich panel.$\lambda_{i}$—The thermal conductivity of one layer in the sandwich panel.$R_{i}$—The thermal resistance of one layer in the sandwich panel.

The thermal loading test was carried out for the purpose of characterizing the fire resistance of the panels. Slight deviations from the standard EN 1363-2 [31] were used. Alternative external fire curves were chosen for the behavioural characteristics of the developed material in different types of fire. These curves are shown in the results. A custom designed furnace was used for the fire testing. Samples with dimensions of 300 mm × 300 mm were placed in a vertical position and were exposed to flames in a direction parallel to the plane of the board. The flame intensity was managed by controlling the flow of gas and was increased over time. Two sensors for temperature monitoring were used to characterize the behaviour of the material in this test. The first was placed in a chamber with a burner and the second on the outside of the flame. The temperature measurement accuracy was 0.1 °C. The temperature was monitored over time.

The number of measured samples for each variant was 30 for density, internal bonding and bending tests. The thermal conductivity was measured on 10 samples for each variant and one sample from each variant was used for the thermal loading test.

### 2.6. Statistical Analysis

The data was characterized using descriptive statistics (sample mean and sample standard deviation) and a multi-factor analysis of variance. The sample standard deviation was calculated according to Formula (3). For the analysis of variance, the following factors were used: lattice count, board density and geopolymer density. Graphically were depicted the influences of the factors on the following variables: bending strength, modulus of elasticity, bending coefficient and internal bonding. Vertical columns in the graphical representation of the analysis of variance represent 95 percent confidence intervals. Limits of the confidence intervals were calculated according to Formula (4). The graphics are listed for illustration of the descriptive statistics. The Tukey HSD test was used to determine if any of the differences between the sample means were statistically significant. A significance level of α = 0.05 was selected. The temperature course during the thermal loading test was depicted using point chart as a function of time.
(3)$$s=\sqrt{\frac{\sum_{i=1}^{N}{\left(x_{i}-\bar{x}\right)}^{2}}{n-1}}$$
where:
*s*—The sample standard deviation.*x*—The observed value.*n*—The number of observations.
(4)$$L_{1,2}=\bar{x}\pm \frac{s}{\sqrt{n}}\times t_{n-1}\left(\alpha \right)$$
where:
$L_{1,2}$—The upper and lower limits of the confidence interval.*s*—The sample standard deviation.$\bar{x}$—The sample mean.*n*—The number of observations.$t_{n-1}\left(\alpha \right)$—The percentile of the t distribution.

## 3. Results and Discussion

Figure 2 shows the effect of the density of the particleboard, the number of lattices and the density of the geopolymer on the bending strength of the sandwich boards. It was found that particleboard density has the greatest impact on bending strength, whereas geopolymer density did not have a statistically significant effect on composite panel properties. In addition, the influence of the inserted lattices was observed where, in the case of the use of one lattice, the bending strength of the test material increased compared to the variants without lattices. With the increasing number of reinforcement elements in the composite, its bending strength generally increases [12]. However, when two reinforcing lattices were used, the bending strength dropped, surprisingly. This seemingly paradoxical phenomenon can be explained by the anchor of the lattices in the composite. Whereas in the case of one-lattice variants, the lattice is firmly anchored in the middle of the particleboard material, for two-lattice variants these lattices are on the surface and are not fully anchored in the composite material. In the variant with two lattices, the lattice is on the borderline of the particleboard and the geopolymer in the neutral zone during bending, and it therefore does not impact the strength of the material [29]. It can be assumed that this lattice located in the neutral zone would affect the impact strength, i.e., a property that is also important in security doors. While the lattice at the bottom edge of the sandwich composite is exactly in the tensile zone during bending and the used lattice has a tensile strength of 600 MPa, due to insufficient anchoring in the particleboard, it did not have a positive effect on the resulting bending strength of the tested material. The geopolymer reference samples with an OSB showed a bending strength of 0.66 MPa for a lower density geopolymer and 0.65 MPa for a higher density geopolymer. The differences between these averages are within statistical error. The reference samples from commercially sold OSB thus achieved a significantly higher bending strength, but this high strength was caused by the significantly higher density of the OSB (620 kg/m^3^) compared to particleboards from rapeseed stalks (340 and 500 kg/m^3^) and a smaller dimension of glued particles [32,33,34].

The developed material exhibited high elasticity, the average modulus of elasticity (MOE) values reached 0.14–0.28 MPa (Figure 3). These are considerably lower MOE values compared to the material on the same rapeseed base [16]; however, in the previous research, another adhesive and another press cycle were used, which implied a different vertical density profile of particleboard. Reference samples with an OSB reached a modulus of elasticity of 0.37 MPa for a lower density geopolymer and 0.29 MPa for a higher density geopolymer. A decrease in MOE was observed along with the decreasing particleboard density used in the sandwich panel, which is consistent with theoretical assumptions [35]. Adversely, the influence of geopolymer density was not observed. When using higher rapeseed particleboard densities, it is apparent (however, not statistically significant) that MOE increases with the number of lattices, but it does not increase at a lower particleboard density. This phenomenon may be due to the anchoring of reinforcing lattices in the board, where these lattices are better embedded in a higher density board and are able to transfer a certain load.

A bending coefficient KbendC (Figure 4) was determined for another characteristic of the bending properties of the developed material. The proportion of material thickness to its minimum bending radius was highest for the sandwich panel with a particleboard density of 340 kg/m^3^ in combination with two lattices and a layer of geopolymer with a density of 300 kg/m^3^. Other differences are not statistically significant at a significance level of 0.05. Compared to wood-based sandwich materials [29,36], the developed material exhibited a lower bending coefficient. However, wood or sandwich material based on lamellae is characterized by high elasticity and a high bending coefficient. The bending coefficient (KbendC) of beechwood is about 0.033 [29]. However, a higher bending coefficient was achieved than in a commercially available wood particleboard (0.01), and values comparable to those of composite material of higher density were obtained, and only from rapeseed particles bonded with epoxy-polyester adhesive [37]. The developed sandwich material thus makes it possible to use boards from rapeseed particles of lower density and bonded with a cheaper, less flexible UF adhesive while maintaining good bending characteristics.

Figure 5 captures the effect of the observed factors on the tensile strength perpendicular to the plane of the boards of the developed sandwich panels. There was a clear influence of particleboard density on the internal bonding of the sandwich composite. With an increasing density of particleboard, the internal bonding of composite panel increases according to theoretical assumptions [32]. On the other hand, the effect of using reinforcing lattices and density of geopolymer on internal bonding was not observed. We can positively assess the fact that there was no breach in the joint between the particleboard and the geopolymer. (As there is no breach of the test specimens in the geopolymer material, this factor is not stated in the chart). The material was breached in the middle of the particleboard at the lowest density point. The lowest particleboard density was attained at its centre thanks to the chosen high-speed closing of the press [38].

The ascertained values of the thermal conductivity coefficient of manufactured panels and the actual measured values of the density of the individual materials are specified in Table 5. Mild variations in the actual density of the particleboard and the geopolymer from their nominal values were found. The effect of reinforcing lattices on thermal insulation properties and on combustion resistance was not assessed. The thermal conductivity coefficient of the sandwich composite with the lowest density value was 0.111 W/(m·K) and just exceeded the value of 0.1 W/(m·K), which is considered a threshold value for thermal insulating materials. According to theoretical assumptions, the highest values of the thermal conductivity coefficient were obtained [39] for the highest density sandwich composite; nevertheless, the value of 0.214 W/(m·K) can be considered an acceptable thermal conductivity value compared to other load-bearing building materials, such as a wall made from wet pine wood [40]. The good thermal insulation properties of geopolymer foam concretes due to air-cavity content were described earlier, the thermal conductivity coefficient ranged from 0.15–0.48 W/(m·K), which is a better thermal insulation compared to the foamed Portland cement concrete of the same density [21]. Low thermal conductivity (0.15–0.4 W/(m·K)) is also described by [41], who prepared geopolymer foams with Al powder as a foaming agent.

Geopolymers can be successfully used to increase the fire resistance of materials [42,43]. Figure 6 captures the burning characteristics of developed panels. The effect of the particleboard and geopolymer density on the fire resistance of the panels was observed. Despite slight variations in temperature rise in the furnace, there was a clear influence of geopolymer density on the resistance time of the panel (the time of reaching the temperature of 100 °C on the outside of the panel). On the contrary, the density of the particleboard layer did not affect its fire resistance as the density of the geopolymer layer. This is the opposite effect of the individual layers, rather than the influence of the individual layers on the mechanical properties where the density of the particleboard is the most important parameter.

## 4. Conclusions

The submitted paper evaluates the physical and mechanical properties of the developed sandwich composite material based on particles of winter rapeseed stalks, geopolymer and reinforcing lattices. The fundamental influence of particleboard density on the resulting mechanical properties of the entire sandwich panel was demonstrated. The density of the second layer of the sandwich panel and the geopolymer did not have the same impact on its mechanical properties as the particleboard density. The reinforcing lattice made of basalt fibre positively influenced the mechanical properties of sandwich composites only if it was sufficiently anchored in the particleboard structure. The developed materials reached a higher bending strength than 0.3 MPa in only two cases, and the tensile strength perpendicular to the board plane was also low. However, these low values are due to the low density of the material and the low adhesive content. On the contrary, good thermal and fire protection properties were achieved, namely the thermal conductivity coefficient of the sandwich composite with the lowest density value was 0.111 W/(m·K) and all developed sandwich composites resisted flame for more than 13 min.

## Figures and Tables

**Figure 1 materials-12-01432-f001:**
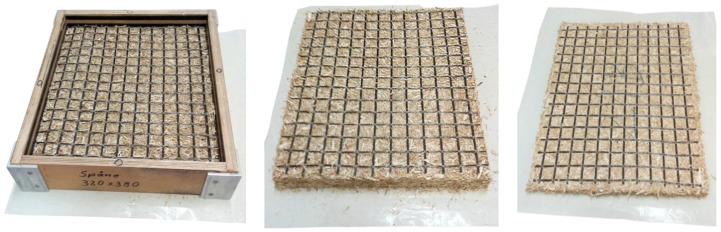
Lattice placed in the surface of the board. From the left: layered carpet in the form for pre-pressing, carpet after cold pre-pressing, pressed board.

**Figure 2 materials-12-01432-f002:**
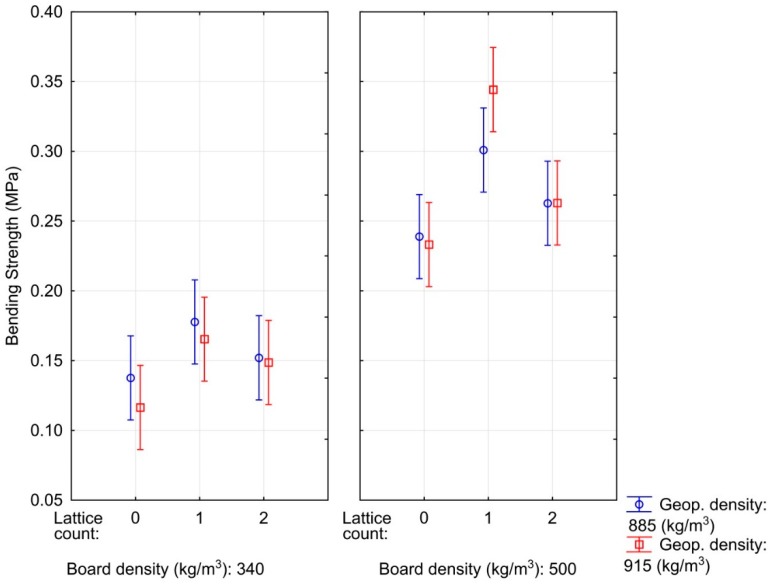
Influence of particleboard density, geopolymer density and the number of lattices on the bending strength of composite materials.

**Figure 3 materials-12-01432-f003:**
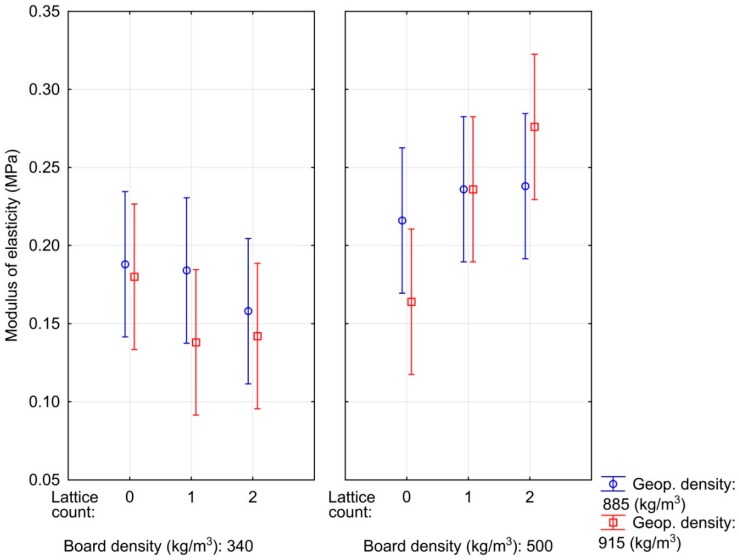
Influence of particleboard density, geopolymer density and the number of lattices on the modulus of elasticity of composite materials.

**Figure 4 materials-12-01432-f004:**
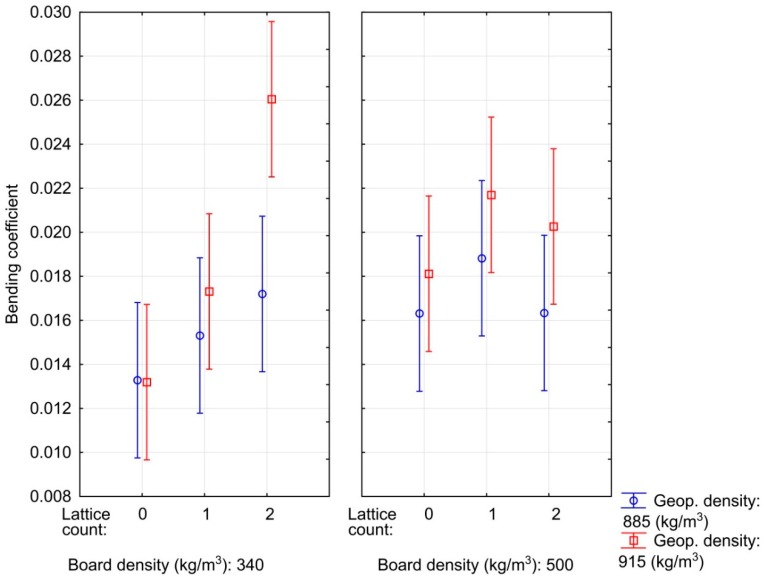
Influence of particleboard density, geopolymer density and the number of lattices on the bending coefficient of composite materials.

**Figure 5 materials-12-01432-f005:**
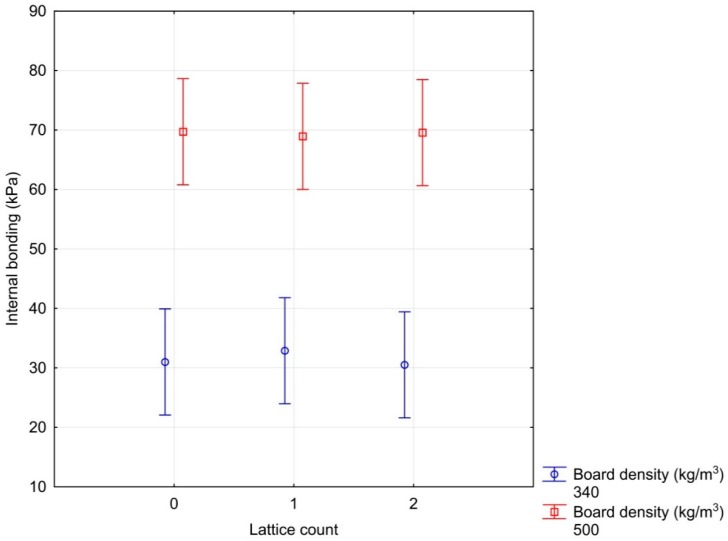
Influence of particleboard density and the number of lattices on the internal bonding of composite materials.

**Figure 6 materials-12-01432-f006:**
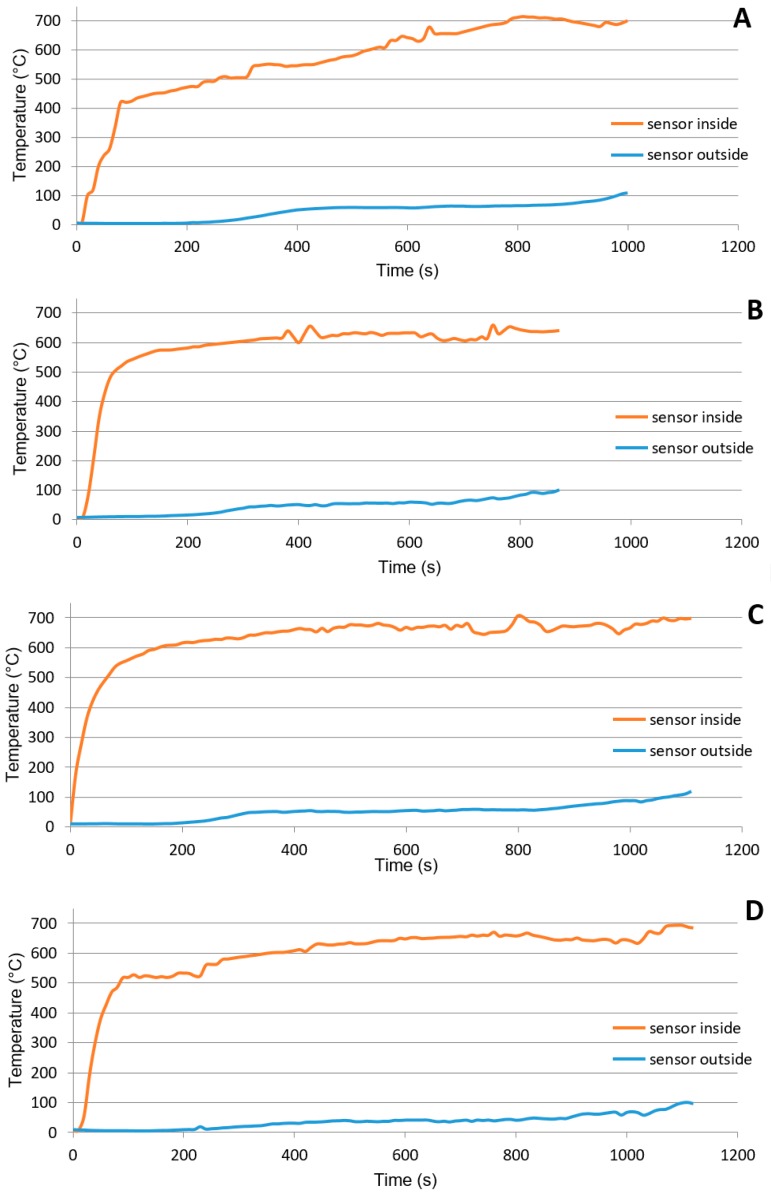
Influence of panel composition on burning characteristics, (**A**) board density 340 kg/m^3^, geop. density 885 kg/m^3^; (**B**) board density 500 kg/m^3^, geop. density 885 kg/m^3^; (**C**) board density 340 kg/m^3^, geop. density 915 kg/m^3^; (**D**) board density 500 kg/m^3^, geop. density 915 kg/m^3^.

**Table 1 materials-12-01432-t001:** Dimensional characteristics of particles.

**Dimension (mm)**	0–0.25	0.25–0.5	0.5–0.8	0.8–1.6	1.6–2	2–3.15	3.15–8
**Percentage (%)**	1.2	2.8	4.8	39.4	20.1	23.1	8.6

**Table 2 materials-12-01432-t002:** Pressing Cycle.

Phase No.	Thickness at the End (mm)	Moving Time (s)	Remaining Time (s)
1	40	0.1	0
2	18	3	0
3	11.8	8	12
4	12	5	10
5	12.3	3	0
6	12	3	141
7	12.5	25	0
8	500	0.1	0

**Table 3 materials-12-01432-t003:** Variants of manufactured sandwich-structured panel.

Layer Specification	Fire-Resistant Sandwich-Structured Panel											
Board density (kg/m^3^)	340						500					
Geopolymer density (kg/m^3^)	885			915			885			915		
Lattice count	0	1	2	0	1	2	0	1	2	0	1	2

**Table 4 materials-12-01432-t004:** Geopolymer composition.

Component	Percentage Share of Individual Components	
	Geopolymer Density 885 kg/m^3^	Geopolymer Density 915 kg/m^3^
Cement Baucis Lk	43.2%	43.4%
Activator Baucis Lk	38.9%	39.1%
Kema Mikrosilika	4.3%	4.3%
Mineral wool Isover	13.0%	13.0%
Aluminium powder	0.6%	0.2%

**Table 5 materials-12-01432-t005:** Average densities of materials and thermal conductivity of boards.

Sandwich Panel Combination	Board Density (kg/m^3^)	Geopolymer Density (kg/m^3^)	λ20/65 (W/(m·K))
1	340 (18)	885 (32)	0.111 (0.009)
2	340 (18)	916 (28)	0.113 (0.014)
3	498 (17)	885 (32)	0.134 (0.008)
4	498 (17)	916 (28)	0.214 (0.013)

Values in parentheses are the standard deviations.

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
