# Peer review of "Fire-Resistant Sandwich-Structured Composite Material Based on Alternative Materials and Its Physical and Mechanical Properties"

_materials, 2019, doi:10.3390/ma12091432_

Round 1
Reviewer 1 Report
Authors MUST remove or acknowledge copied text.
Figure 2 needs improving - axis, definition of specific pressure, reduce in size, caption.
Formatting of units (especially density superscripts).
Be consistent with capitilisation (eg table vs Table).
Authors should explain why an unsteady method was chosen for measurement of the thermal conductivity.
Authors should give details of experimental uncertainty and how the error bars were calculated.
Figures 3, 4, 5: improve axes labels (x axis looks messy, use dot instead of comma in english)
The authors have not adequately described the fire tests (they say deviations from the standard were used but they do not specify what these were).
Additionally see comments in attached file.

Author Response
Dear Reviewer 1, thank you for your constructive comments and hints how to enhance the paper. Please find enclosed response to your comments and revised manuscript.

Reviewer 2 Report
The literature background is insufficient, you need a more detailed and focused review, functional to the message and conclusions of your work.
No research significance was found from your paper. A comparative study with the existing filling materials to show this sandwich structure composite processing excellent performance with very low price will be more persuasive.
Author Response
Dear Reviewer 2, thank you for your comments. Please find enclosed response to your comments and revised manuscript.

Round 2
Reviewer 1 Report
The first paragraph of section 2 has had important information removed: rape stalk supplier, method of measurement (screen analysis); which should be retained.
Section 2.5 - the authors have not specified how long the samples were cured for, as requested
The statistical analysis is still not adequately described.
Eq 1 is still unclear - is it a: h/(I/12y) or b: (h/I)/12y ?
Author Response
Dear Reviewer 1, thank you for your comments. Please find enclosed response to your comments and revised manuscript.

Reviewer 2 Report
Agree to publish
Author Response
Dear Editor, since Reviewer 2 submitted positive statement, we do not have any comments.
Round 3
Reviewer 1 Report
In the description of the experiment and the statistics, the authors must provide:
- details of how the standard deviations and confidence limits were calculated (number of samples, how the confidence limits were calculated)
- details of experimental errors, and how these influence the accuracy of the results
- how the Tukey test was used to establish significance (it is not adequate to simply say 'the Tukey test was used' - sufficient details must be given for readers to perform their own replication of the analysis)
Author Response

(The authors gave the same response as above.)
